# Effect of an Educational Nursing Intervention on the Mental Adjustment of Patients with Chronic Arterial Hypertension: An Interventional Study

**DOI:** 10.3390/ijerph19010170

**Published:** 2021-12-24

**Authors:** Ana Margarida Alves, Alexandre Rodrigues, Pedro Sa-Couto, João Lindo Simões

**Affiliations:** 1Inpatient Service of Surgical Specialties, Centro Hospitalar do Baixo Vouga E.P.E., 3810-164 Aveiro, Portugal; amargaridamartins93@gmail.com; 2School of Health Sciences (ESSUA), University of Aveiro, 3810-193 Aveiro, Portugal; alexandre.rodrigues@ua.pt; 3Centre for Innovative Biomedicine and Biotechnology (CIBB), University of Coimbra, 3004-531 Coimbra, Portugal; 4Center for Health Studies and Research, University of Coimbra, 3004-531 Coimbra, Portugal; 5Centre for Research and Development in Mathematics and Applications (CIDMA), Department of Mathematics (DMAT), University of Aveiro, 3810-193 Aveiro, Portugal; p.sa.couto@ua.pt; 6Institute of Biomedicine (iBiMED), University of Aveiro, 3810-193 Aveiro, Portugal

**Keywords:** chronic disease, hypertension, nursing, patient education, mental adjustment

## Abstract

The objective of this analytical and interventional prospective quantitative study was to assess the effect of an educational intervention performed by nurses for mental adjustment to chronic disease in patients with hypertension. A convenience sample was studied, composed of 329 participants with chronic hypertension, followed in a primary healthcare unit in the Central Region of Portugal. Data collection was carried out by applying the Mental Adjustment to Disease Scale (MADS) before and 1 month after the educational nursing intervention between September 2017 and February 2018. Prior to the application of the educational intervention, 43.5% of the participants were classified as “unadjusted” in at least one of the subscales of MADS. After the educational intervention, 21.3% of the participants classified as “unadjusted” became “adjusted” in all MADS subscales. The success rate of the intervention varied from 26.9% (in the fatalism subscale) to 44.6% (for the anxious concern subscale). Participants were more likely to be mentally “unadjusted” to hypertension if they lived with other family members, had an active professional situation before the diagnosis of hypertension, still had an active professional situation now, were under 65 years old, had a shorter time to diagnosis (1–2 years), and measured blood pressure less regularly. The educational intervention performed by nurses is relevant for the mental adjustment of hypertensive patients, contributing to increased knowledge, as well as improvement in preventive and self-care practices, facilitating the experience of the health/disease transition process.

## 1. Introduction

According to the World Health Organization, hypertension is one of the world’s leading causes of death and affects nearly 1.28 billion adults aged 30–79 years worldwide [1]. According to Yildiz and Erci, this pathology is responsible for the deaths of 7.6 million people a year (54% of strokes and 47% of ischemic heart diseases) and is diagnosed in 90 million people every year [2]. As far as chronic diseases are concerned, cardiovascular diseases are the leading cause of death in the Portuguese population [3].

Hypertension is a relevant risk factor for cardiovascular diseases, which may result in myocardial infarction, stroke, congestive heart failure, and renal failure, among others [4]. As a pathology that can be prevented with changes in lifestyles, the adoption of control and self-care measures is a major factor for a balanced health/disease transition and for an appropriate mental adjustment to the disease.

Since prevalence of hypertension increases with age, the management of hypertension and the prevention and treatment of important complications related to hypertension will continue to be a global challenge for health professionals [4].

Family nurses (FNs) are a fundamental link in this transition process, and they can help individuals with hypertension and their families to adhere to the appropriate therapeutic and medication regimen [5]. Thus, the role of nurses in promoting self-care includes planning, execution, and evaluation of nursing interventions, investing in the stimulation of the individual to change lifestyles, increasing awareness of potential complications of hypertension, and observing behavioral changes after such educational intervention [2].

Educational health promotion interventions contribute to increasing people’s knowledge about hypertension and having a positive influence on their beliefs and motivation to self-engage in the process of adaptation to the disease [6]. Thus, educational interventions can create opportunities for patients to better understand their conditions and the importance of adhering to health promotion measures or disease control, as well as to raise awareness about the progression of the disease and its possible complications. The implementation of an effective educational intervention may also contribute to mental adjustment to arterial hypertension, although there are no studies addressing this problem. Considering the studies carried out in patients with other chronic diseases, nurses should promote self-care through an educational intervention, including the assessment of the person’s needs, the planning of strategies adapted to the needs, and training for self-management of chronic disease [7,8].

## 2. Background

Over the past few years, technological and scientific advances have contributed to a significant increase in average life expectancy [9]. However, combined with longevity, the adoption of unhealthy lifestyles has contributed to an exponential increase in chronic pathologies and, consequently, new challenges are posed: the need to experience a health/disease process, to reorganize, and to adapt. Thus, considering the above, in the context of chronic pathologies, cardiovascular diseases are the main cause of mortality in the Portuguese population [10]. Arterial hypertension has gained relevance, due to its high prevalence and to the fact that published epidemiological studies demonstrate insufficient control, which may be reflected in the abnormal incidence of stroke [11].

According to Williams et al., about 40% of adults over 25 have hypertension worldwide [12]. For the past 30 years, Portugal has been described as one of the countries with high levels of mean arterial pressure. Thus, the results of the Portuguese Society of Hypertension show that the prevalence of hypertension in Portugal is 40.2% [3].

Arterial blood pressure depends on the balance between numerous mechanisms and, most of the times, it is not possible to determine the cause of hypertension. However, there are a small number of clinical cases where it is possible to clearly delimit the origin. Hypertension is diagnosed in two ways: primary and secondary. Primary hypertension occurs in about 90% to 95% of cases and may be due to environmental factors (salt intake or stress), peripheral insulin resistance, obesity, and the renin–angiotensin-aldosterone system; the simultaneity of two or more risk factors (such as overweight, high salt intake, tobacco, alcohol, and stress) and hereditary predisposition may also be considered. In secondary hypertension, which occurs in about 5% to 10% of cases, it is possible to identify a responsible pathology, such as kidney disease, endocrine or catecholamine alterations, aortic artery malformation, or neurological problems such as brain tumors, brain hemorrhages, and head trauma. There are also cases associated with the prescription of certain drugs and ingestion of toxics, such as lead and mercury [13].

In addition to chronic disease, hypertension is considered one of the main public health problems that can coexist with other cardiovascular risk factors, such as tobacco consumption, diabetes mellitus, dyslipidemia, sedentary lifestyle, and obesity, which further aggravate cardiovascular risk and consequently increase the risk of morbidity and mortality [14]. Despite being initially asymptomatic, it may present complications at the cardiac, cerebral, ocular, and eventually renal level, which may lead to chronic renal failure. It is a prolonged illness, often associated with a certain degree of disability, making patients more vulnerable to the risks that can affect their health [15]. Therefore, these complications and low treatment adherence put the life of the patient with hypertension at risk, with sequels remaining with a strong negative impact on quality of life.

After the diagnosis of a chronic disease such as this one, both the patient and the family experience a crisis, which, according to Martins, Monteiro, and Gonçalves, may be motivated by the emotional stress associated with the situation of serious illness, the treatment, symptoms, and chronicity, together with progressive deterioration, and facing the possibility of death [16]. On the other hand, according to Rodrigues, Ferré-Grau, and Ferreira, it could be motivated by responsibility for physical care to the patient, with all the necessary implications for daily, family, and professional routines [17]. Due to its permanent nature and irreversible changes, it challenges the development of knowledge and skills for the management of the disease and its therapeutic regimen, with the aim of controlling its evolution, preserving autonomy [18].

In this context and according to Afonso, most patients and families easily recognize the disease’s physical effects and impacts, but the perception of emotional sequels is less frequent and less accepted [19]. Emotional responses to the disease vary according to the different stages of the disease and treatment, and the two greatest responses are fear and suffering. This fear is related to the fear of dying, losing vital functions, losing affections and approval, pain, and the unknown. Denial derives from fear, and various defense mechanisms are activated to protect the patient from emotional disorganization, making it possible to dedramatize some threats of the disease situation. This attitude protects patients by providing them with the time needed to assimilate the impact of the disease, and its use gradually decreases as it faces the threats posed by the situation. It is never completely abandoned, remaining to a reduced degree for the preservation of hope for a satisfactorily normal and productive life. Thus, the prolongation of the state of denial as far as chronic disease is concerned indicates that the patient has inadequate internal resources to deal with the threats imposed.

Given these implications for the patient and their family, mental adjustment becomes a fundamental individual tool to respond to any chronic disease, which, according to Santos, Ribeiro, and Lopes, can be defined as the set of cognitive and behavioral responses expressed by an individual in the face of diagnosis [20]. According to the same authors, this adjustment may have more or less adaptive responses. The so-called “fighting spirit” is seen as the most adaptive set of responses, while “despair/hopelessness” are considered as unadjusted responses. “Fighting spirit” is the set of coping responses of confronting type, manifested by an optimistic vision of the future and belief in the possibility of some control over the disease. On the other hand, “despair/hopelessness” refers to the set of passive responses to the disease, expressed essentially by a negativistic view of its consequences, as well as not believing in the possibility of any control.

The process of adaptation is a fundamental condition, and nonadaptation is a complex phenomenon because the latter requires active participation of the person in treatment, and changing lifestyle may be needed to obtain the maximum benefit of it [21].

Burnier and Egan systematized a set of non-adhering factors associated with hypertension, which can be grouped into four large dimensions: factors related to the patient, factors related to patient/professional relationship, factors related to the disease and treatment, and factors related to the social and institutional context [22]. Therefore, it is possible to establish a relationship between the treatment regimen and mental adjustment.

In this sense, health professionals in general and FNs in particular have skills that enable them to support the patient and their family in the process of adhering and, consequently, of better adjusting to the disease. Their intervention can be at a lifestyle modification level after diagnosis of hypertension, adjusted to personal characteristics and to interaction with the environment. The inability of patients to adjust to the disease can lead to negative effects, such as nonadherence to treatment or the perception of decreased quality of life.

It is up to nurses to train patients with hypertension and their families, providing information about the therapeutic regimen, whilst they motivate and stimulate them to become proactive. Strategies aimed at implementing learning teaching processes are used so they can acquire skills that enable them to develop behaviors appropriate to their health [11].

Hence, the surveillance nursing consultation of hypertension is essential, and FNs implement this to establish partnerships with patients and their families. Their goal is to promote treatment adherence making information available, thus improving their health literacy. Furthermore, proper use of medicines is considered to avoid cardiocerebrovascular complications, and links between different levels of the health system are established so that patient can properly manage the disease according to their level of risk [23].

The intervention of the FN should be performed to provoke changes in the cognitive, affective, and behavioral domain of the patient and the family. Jiang et al. suggested that the intervention programs of these health professionals promote awareness and knowledge about the effects of cardiovascular diseases; therefore, the behaviors of patients will eventually be modified [18].

The aim of the study was to assess the effect of an educational tool used by nurses for mental adjustment to chronic disease in patients with hypertension, followed in a primary healthcare unit and enrolled in the hypertension program information system used in Portugal.

## 3. Materials and Methods

### 3.1. Design

An analytical prospective study was conducted in which the efficiency of an educational tool for mental adjustment to chronic arterial hypertension was tested, and significant relationships were established between the variables under study. Therefore, it is an interventional study where the effect of a standardized intervention was assessed in the promotion of mental adjustment in participants who were not mentally adjusted in at least one of the subscales of the Mental Adjustment to Disease Scale (MADS).

### 3.2. Sample/Participants

A convenience sample of people with chronic hypertension followed in a primary healthcare unit in the Central Region of Portugal was studied. The inclusion criteria were age equal to or greater than 18 years, being enrolled in the hypertension program of the information system “SClínico” (“SClínico | primary healthcare (CSP)—SPMS”) for more than 1 year, and attending surveillance nursing consultations in the health unit. Exclusion criteria were not mastering Portuguese language (important for the educational intervention to be applied), presenting diagnosed psychiatric alterations (according to diagnoses presented in the information system “SClínico”), or not having cognitive ability to understand the questions (score in the Mini Mental State Examination (MMSE) less than 22) [24].

To determine the sample size, the Raosoft Sample Size Calculator (Raosoft Inc., Seattle, WA 98115, USA) tool was used, referring to a margin of error of 5% and a 95% confidence level. The number of patients enrolled in the healthcare unit where the study was conducted was 12,473, and the number of patients enrolled in the hypertension program of the “SClínico” information system was 2240 (study population). After calculating the recommended sample size, a value of at least 333 participants was obtained.

Thus, the final sample of our study consisted of 329 participants. At the first stage of psychological adjustment evaluation, 333 people were assessed, but four were excluded because they presented an MMSE score below 22. At the second stage, 141 people were assessed (143 participants who were classified as “unadjusted” at the first stage, minus two patients who did not consent to participate at the second stage of data collection) and were the object of the educational intervention referred to above (see Figure 1).

### 3.3. Data Collection

Data collection was performed at two stages: before and 1 month after the educational intervention, considering patients’ self-assessment (changes that occurred, from a perspective of adaptability and production of new knowledge). These evaluations took place between September 2017 and February 2018.

Initially, 329 participants answered a questionnaire that was fulfilled by the researcher, since most of the participants were elderly and with low literacy. The data collection instrument consisted of a first section with questions related to sociodemographic and clinical characterization and a second section with items related to the MADS.

Thus, the first stage included variables such as age, gender, marital status, academic qualifications, professional situation (prior to the disease and currently), number of members and composition of the household, duration of illness, taking of medication to treat hypertension, going to hypertension surveillance consultations, site where blood pressure is checked, and how often blood pressure is measured.

At the second stage, the MADS enabled the evaluation of a set of cognitive and behavioral responses to the diagnosis of hypertension, to determine the mental adjustment to this chronic disease. This scale was adapted and validated by Sá [25] for the Portuguese population and consists of 47 items, divided into five areas:-*Fighting spirit* (items 4, 5, 6, 11, 13, 16, 18, 20, 26, 27, 28, 31, 32, 34, 39, 40, 41, 42, 43, 44, 46, and 47) that determines the existence of a set of coping responses, optimistic view of the future, and belief in the possibility of some control over the disease;-*Despair/hopelessness* (items 2, 9, 17, 23, 25, and 36) that delimits the existence of passive responses to the disease, expressed essentially by a negativistic view of competences, as well as not believing in the possibility of any control over it;-*Anxious concern* (items 1, 3, 10, 14, 19, 21, 22, 29, and 37) that establishes the existence of persistent anxiety, sometimes accompanied by depression, as well as compulsive search for information as a behavioral response about the disease, but tendency to interpret it in a pessimistic way, being uncertain the ability to control the disease;-*Fatalism* (items 7, 8, 12, 15, 24, 30, 33, 35, and 45) that enables us to know if the patient sees the diagnosis as a minor threat with absent attitudes of confrontation and serenity about the results—passive acceptance;-*Avoidance/denial* (items 38) understood as the refusal of diagnosis or admission of diagnosis but denying the severity with a positive view of the prognosis in which the issue of control is irrelevant.

In each question, there are four answer options, counted from 1–4 points: “It does not apply at all to me” (1), “It does not apply to me” (2), “It applies to me” (3), and “It applies totally to me” (4). The total score represents the sum of the questions of each area.

After the first stage of data collection, the score of each dimension was calculated, and then the “adjusted” or “unadjusted” classification of each participant in each dimension was established, as indicated in Table 1. Afterward, a standardized educational intervention was planned, adapted to each participant, according to the subscales where they were “unadjusted”. In this intervention, the following topics were addressed: medication regimen, diet, physical exercise, hypertension as a chronic disease, complications of hypertension, self-monitoring of blood pressure, correct measurement of blood pressure, quality of life in hypertension, and/or more frequent symptoms in hypertension. These topics were grouped according to the specificity of each subscale, as can be seen in Table 2.

The second stage of data collection was applied at least 1 month after the first evaluation to participants classified as “unadjusted” in at least one of the subscales and who were submitted to educational intervention.

### 3.4. Validity and Reliability/Rigor

The MADS was adapted and validated by Sá [25] and was based on the Mental Adjustment to Cancer Scale [26]. This scale already underwent further updates for the oncology area [20,27]. With the adaptation and validation performed by Sá [25] the scale can now be used in any chronic disease in addition to cancer and, therefore, the word “cancer” was replaced by “disease”.

As this scale was never used in hypertensive patients, to ensure its internal consistency, the Cronbach’s alpha calculation was performed. Alpha values are typically rated as excellent (if greater than or equal to 0.9), good (if between 0.8 and 0.9), acceptable (if between 0.7 and 0.8), questionable (if between 0.6 and 0.7), poor (if between 0.5 and 0.6), and unacceptable (if less than 0.5). Table 3 shows the results obtained and their comparison with those of other studies [25,26,28,29]. As can be seen, Cronbach’s alpha values were in line or higher than the values reported in the literature, except for the “fighting spirit” subscale. In this subscale, the alpha value is considered unacceptable. The analysis of internal consistency led us to the alpha value of 0.611 if we eliminated seven items (items 5, 20, 27, 43, 45, and 46). The removal of the remaining items did not increase the Cronbach’s alpha value. However, as our goal was not the construction of a new subscale, we chose to use the original “fighting spirit” subscale, i.e., with the 22 items. We return to this aspect in the limitations of the study.

### 3.5. Ethical Considerations

The study was authorized by all management bodies of the primary healthcare unit after the assent of the Ethics Committee of the Regional Health Administration of the Center (File No. 43/2016). The use of the scales applied in the study was also authorized by their authors.

From all participants, free and informed consent was obtained after being invited to participate in the study, explained the objectives, and guaranteed anonymity and confidentiality of the data, through the attribution of a sequential numerical code to each participant, which was used to identify the participants at the two stages of evaluation.

### 3.6. Data Analysis

Data analysis was performed using the Statistical Package for the Social Science (SPSS) version 24.0 program for Windows (Microsoft Corporation, Washington, WA, USA). For the analysis of sociodemographic and clinical data, descriptive statistics were used: absolute frequencies (*n*), percentages (%), means (M), and standard deviations (SD). The characterization of mental adjustment was made according to each of the MADS subscales presented (fighting spirit, despair/hopelessness, anxious concern, fatalism, and avoidance/denial) and according to sociodemographic and clinical variables.

To identify possible relationships between the variables studied and the proposed classification of “adjusted” versus “unadjusted”, the odds ratio (OR) and its respective 95% confidence interval (95% CI) were calculated through a univariable binary logistic regression, for the different MADS subscales used.

To determine whether the intervention had the desired effect, the proportion of successes and their respective confidence interval of 95% were calculated for participants classified as “adjusted” at stage 2. Note that success represents a participant classified as “unadjusted” at stage 1 and as “adjusted” at stage 2. Results were classified as significant if the *p*-value was less than 0.05.

## 4. Results

### 4.1. Sociodemographic and Clinical Characteristics of Participants

In relation to gender, 41.3% of the participants were male and 58.7% were female (see Table 4). Regarding marital status, the categories presented show unequal class differences, since 63.5% were married/living with a partner and only 3.6% were divorced/separated. The study sample was generally schooled with only 10.6% without schooling. When analyzing the composition of the household, it was possible to verify that most of the participants lived with a companion, considering that 62.9% lived with a spouse/partner and 28.3% lived alone, living on average with 2.1 people. As for profession, 61.7% of the participants were nonskilled workers. Regarding age, participants were on average 70.8 years old with a standard deviation of 11; after establishing the age groups, it was possible to notice that 12.2% were between 51 and 60 years old, 28.9% were between 61 and 70 years old, and 32.5% were between 71 and 80 years old.

It was confirmed that most participants were retired (52.6%), although there was a relevant percentage of participants (45.6%) who were employed prior to diagnosis. At the time of the study, the prevailing professional situation was retired (78.1%).

Regarding clinical characterization of the participants, it was found that 99.7% of them took medication for hypertension and 93.3% were doing it for more than 1 year (see Table 5). As for the frequency of surveillance consultations of hypertension, 95.7% reported going to all surveillance consultations, and the main reason for missing them was forgetfulness (78.6%). Checking blood pressure is essential in the control of the pathology, and the results show that 35.6% took blood pressure once a month and 25.5% once a week, and the measurement was performed in consultations and at home (47.7%). On average, the diagnosis was made 8.4 years ago, and most participants were diagnosed with this pathology in the period of 3–10 years (3–5 years: 27.4% and 6–10 years: 27.4%).

### 4.2. Specificities of Mental Adjustment in Patients with Hypertension

For a better understanding of the behavior of the mental adjustment subscales, some relationships were established with the sociodemographic variables, which are reported below.

Regarding the evaluation of the despair/hopelessness subscale (see Table 6) at stage 1, statistically significant results were found in the following variables: “household”, “professional situation prior to the disease”, and “current professional situation”. Thus, patients living with two persons (OR = 2.30; 95% CI = 1.05–5.06), and who were employed/unemployed prior to the disease (OR = 2.55; 95% CI = 1.48–4.40) and currently (OR = 2.59; 95% CI = 1.46–4.62), were more likely to be “unadjusted” in this subscale than those who lived alone, and who were retired before the disease and currently, respectively. At stage 2, no statistically significant results were found for all variables studied.

In the anxious concern subscale (see Table 7) at stage 1, statistically significant results for the “unadjusted” category were found in the variables “academic qualifications”, “age”, “household”, “professional situation prior to the disease”, “current professional situation”, “duration of disease”, and “frequency of blood pressure measurement”. Thus, patients with secondary/higher education (OR = 5.93; 95% CI = 1.44–24.36), aged less than 65 years (OR = 3.36; 95% CI = 1.51–7.50), living with 2 (OR = 1.74; 95% CI = 0.71–4.23) or more persons (OR = 3.92; 95% CI = 1.61–9.54), employed/unemployed before disease (OR = 2.38; 95% CI = 1.31–4.31) and currently (OR = 3.74; 95% CI = 2.03–6.89), with disease duration between 1 and 2 years (OR = 2.99; 95% CI = 1.27–7.00), and who checked blood pressure once to three times a year (OR = 2.62; 95% CI = 1.26–5.48) were more likely to be “unadjusted” to this subscale. At stage 2, statistically significant results were only found in the “household” variable. In this case, the results were contrary to those at stage 1. At stage 2, users living alone (OR = 9.90; 95% CI = 1.53–63.69) were more likely to be “unadjusted” in this subscale than those living with other persons.

Regarding the fatalism subscale (see Table 8) at stage 1, statistically significant results were found in the variables “professional situation prior to the disease” and “frequency of blood pressure measurement”. Thus, patients who were employed/unemployed before the disease (OR = 1.73; 95% CI = 1.00–2.97) and who checked blood pressure once to three times a year (OR = 2.53; 95% CI = 1.31–4.91) were more likely to be “unadjusted” in this subscale than patients who were retired, and who took blood pressure every day or once a week, respectively. At stage 2, no statistically significant results were found.

In relation to the avoidance/denial subscale (see Table 9) at stage 1, statistically significant results were found for the “current professional situation” variable. In this sense, employed/unemployed users at the time of the study (OR = 2.00; 95% CI = 1.03–3.92) were more likely to be “unadjusted” in this subscale than retired users. Furthermore, there were no statistically significant results at stage 2 in this subscale.

### 4.3. Results of the Educational Intervention in the “Unadjusted” Participants

To achieve the main objective of this study, it was necessary to classify participants as “adjusted” and “unadjusted” at stages 1 and 2 of data collection. Thus, in Table 10, we can see, by subscale, the respective classification at these two stages.

At stage 1, a total of 329 participants were assessed; about 56.5% (*n* = 186) of participants were classified as “adjusted” in all subscales of MADS, while 43.5% (*n* = 143) were classified as “unadjusted” in at least one of these subscales. However, the total number of “unadjusted” classifications was 244, which indicates that the same participant was classified as “unadjusted” in multiple subscales; 22.5% (*n* = 74) were classified as “unadjusted” in two subscales simultaneously, 5.1% (*n* = 17) were classified as “unadjusted” into three subscales simultaneously, and 3% (*n* = 10) were classified as “unadjusted” in the four subscales simultaneously.

At stage 2, a total of 98.6% (*n* = 141) of participants were assessed (two patients refused to answer at the second stage); 21.3% (*n* = 30) were classified as “adjusted” in all subscales of MADS, while 78.7% (*n* = 111) remained classified as “unadjusted” in at least one of the subscales studied. The total number of “unadjusted” ratings was 154, indicating again that the same participant was classified as “unadjusted” in multiple subscales; 23.4% (*n* = 33) were classified as “unadjusted” in two subscales simultaneously, 5.6% (*n* = 8) were classified as “unadjusted” into three subscales simultaneously, and only 1.4% (*n* = 2) were classified as “unadjusted” in the four subscales simultaneously. As can be seen, the intervention had a positive effect, either via a decrease in the number of “unadjusted” or a decrease in patients classified as “unadjusted” in multiple subscales. All these patients were classified as “unadjusted” at stage 1.

After the comparative analysis of the results obtained at stages 1 and 2, the effectiveness of the intervention could be verified, since the obtained results show, for each subscale and each moment, participants classified as “adjusted” and “unadjusted” (see Table 10).

It is noteworthy that, regarding the fighting spirit subscale, all participants (329) were classified as “adjusted” at stage 1; hence, they did not require intervention at this level and were not assessed at stage 2. In the remaining subscales, individuals were classified into both groups.

For the despair/hopelessness subscale, 33.8% (95% CI = 22.5%–45.1%) of the participants classified as “unadjusted” at stage 1 were classified as “adjusted” at the second stage (note that one initial participant refused to answer at stage 2), indicating the success rate of the intervention for this subscale. Of note, the mean values for both stages were quite similar for the categories of “adjusted” and “unadjusted”.

Regarding the anxious concern subscale, 44.6% (95% CI = 31.2%–58.1%) of the participants classified as “unadjusted” at stage 1 were classified as “adjusted” at the second stage (as in the previous subscale, one participant refused to participate at this second stage of evaluation). The mean values for the “adjusted” category presented a higher value for stage 2 when compared to stage 1, still reflecting a more anxious concern attitude.

Regarding the fatalism subscale, the success rate of the intervention was 26.9% (95% CI = 16.0%–37.8%), and, for the avoidance/denial subscale, the success rate was 42.6% (95% CI = 27.8%–57.2%). For the despair/hopelessness subscale, the mean values for both stages were quite similar for the categories of “adjusted” and “unadjusted”.

## 5. Discussion

The sociodemographic characteristics of the participants of this study are in agreement with the findings of different studies regarding gender [30,31,32], average ages [31], schooling [30,31], and household [2].

The clinical aspects analyzed demonstrate the high percentage of patients with a prescription for this pathology; thus, both in this study and in the study by Teh et al., almost all individuals had two or more antihypertensive drugs prescribed [33]. Furthermore, it is notorious that medication time is extended [34] and usually accompanies the time to diagnosis [30]. In a study carried out in Portugal with the aims to characterize the medication consumption profile and explore the relationship of beliefs and daily medication management on medication adherence by home-dwelling polymedicated elderly people, the authors found that group C “cardiovascular system” drugs (Anatomical Therapeutic Classification code, using the WHO Collaborating Center for Drug Statistics Methodology’s web site) was the second most representative [35,36].

Health surveillance of these patients is a central aspect in monitoring their overall health status and the progression of this pathology in particular. Blood pressure monitoring is used as an indicator of the outcome of the therapeutic measures implemented and of adherence to salutogenic behaviors. On the other hand, surveillance consultations are also extremely important in the control of signs and symptoms and in the implementation of an intervention targeting them [37].

Regarding the influence of sociodemographic and clinical variables on the mental adjustment of patients with hypertension, it was found that, prior to the educational intervention, participants who lived with other family members and who were at working age at a professional level were more likely to be “unadjusted” in the despair/hopelessness subscale than those who lived alone and were retired. Ojike et al. found in their study that about 3.2% of the hypertensive participants studied were considered psychologically distressed, with higher rates in women and black people [31]. In a cross-sectional study conducted in nine European countries (Armenia, Azerbaijan, Belarus, Georgia, Kazakhstan, Kyrgyzstan, Moldova, Russia, and Ukraine), the authors found a significant association between psychological distress and hypertension; hypertensive patients were more likely to have symptoms of despair or distress (OR = 2.27 (95% CI = 1.91–2.70)) [38].

Regarding the results related to the influence of the household on mental adjustment, the literature states that family is the basic social unit of the general population and may have an important effect on mental health at all ages [16]. On a day-to-day basis, patients with diseases such as diabetes, cardiovascular disease, and asthma are advised to take medications at complex times, maintain special diets, be physically active, self-monitor regularly, and respond to changes in their symptoms and results obtained. Given the complexity of these tasks, many patients need support between health consultations to manage their disease successfully [39]. Thus, family members are increasingly recognized as important allies in the care of chronically ill patients and may also be an important resource for patients not to feel desperate. However, in our study, the results were contrary to this evidence, as patients who lived with two persons were those who were unadjusted in the despair/hopelessness subscale.

According to the results obtained, it was found that patients at working age were the ones who were more unadjusted. This result was also surprising. Retirement is a well-known risk factor for mental health problems, while returning or getting work is a protective factor [40]. According to Nazarov et al., retirement is a loss at the social level, with a major impact on the lives of individuals and families, and it is a transition that involves gains and losses, while the result depends heavily on personality and other individual circumstances [41]. Therefore, it seems plausible to us that retired people, due to the loss of their working and relational activity, may develop feelings of despair, which was not verified in our study. Moreover, in relation to our results, they seem surprising to us because the fact that hypertension does not produce “visible” symptoms and people are more concerned about employment or unemployment than about the situation of hypertension could lead to a better adjustment, which we did not verify in our study.

Regarding the anxious concern subscale, prior to the educational intervention, participants with more schooling, who were younger, who lived with other family members, who were at an active age, with shorter time to diagnosis, and who measured blood pressure with less often had a greater possibility of presenting “unadjustment” in this subscale.

Like patients with other chronic pathologies, hypertensive patients experience many deep emotions that increase the risk of developing mental disorders, especially anxiety and depression [30,42]. In the study by Kretchy et al., the hypertensive patients studied had symptoms of anxiety (56%), stress (20%), and depression (4%) [30]. These authors also verified that stress increased the probability of nonadherence to therapy (OR = 2.42 (95% CI = 1.06–5.5), *p* = 0.035).

According to Spruill, exposure to chronic stress has been hypothesized as a risk factor for hypertension, and occupational stress, including work-related stress, stressful aspects of the social environment, and low socioeconomic status have each been studied extensively [43]. Most adults spend a substantial portion of their lives at work; as such, it should not be surprising that chronic job stress can have a powerful impact on health. Thus, it seems plausible that participants who have an active life may be mentally more unadjusted by these factors favoring stress.

A low level of literacy is associated with worse outcomes in treatment adherence, including low knowledge of health, increased incidence of chronic diseases, intermediate markers of poorer diseases, and insufficient use of health promotion services [44]. In this sense, it is understandable that patients with higher literacy levels and, therefore, with greater health knowledges are more concerned about their health situation, demonstrate more anxiety, and consequently are more mentally unadjusted to the disease. This fact may be related to access to information, as patients with more information understand better the consequences of uncontrolled hypertension and may develop more anxiety.

Quality of life in chronic diseases may vary with age, especially for older adults. Chronic diseases affect the mobility of the elderly and, consequently, their physical and functional state. Moreover, emotional balance and self-esteem decrease due to dependence on older people [45]. These diseases contribute to a reduction in the quality of life of the elderly and are associated with unhappiness and psychical suffering, resulting in poor quality of life [46]. However, the results of this study show that younger adults have a higher possibility of presenting symptoms related to anxiety, and they find it more difficult to overcome challenges. Certain modifiable factors may mediate the associations between stress and mental health, including sleep, loneliness, and resilience, which could be at the origin of our results.

As seen above, previous studies indicated that family and work activity are considered important factors for mental adjustment, in this case, for a reduction in anxiety-related symptoms. In this regard, Ojike et al. found that retired people, because of the loss of their professional activities, present psychological changes more frequently, such as anxiety and depression [31]. Regarding family, according to Bell et al., many elderly people, in addition to physical limitations, have sensory and/or cognitive limitations that limit their ability to hear, see, understand their health circumstances, or manage their chronic disease process, thus developing anxiety symptoms related to this disability [47]. Therefore, family support in this process is fundamental and is a very important factor in reducing anxiety of the patients with hypertension [48]. However, in our study, the results were contrary to the evidence found that those who live with other family members or who are younger develop more anxiety. This situation is perhaps related to the fact that, in larger families, it is more difficult to manage relationships, and younger people are more concerned about the situation of probable unemployment than older people.

Another important factor determining a patient’s adjustment to chronic disease is the acceptance of their illness. This is a complex psychological phenomenon, which is of a constructive significance [49]. In the present study, the time of diagnosis was another factor that influenced “anxious concern”. Thus, it was found that patients with less time to diagnosis had more symptoms related to anxiety. We think that this result is related to the short time of diagnosis and because patients have not yet accepted their disease, developing more symptoms of anxiety. Acceptance of an illness is a process that consists of many stages, such as shock, confrontation, escape, and assimilation, and that depends on many factors. Everyone is different; hence, the reaction to the disease and acceptance of the new situation are also different. In the study of Pluta et al., statistical analysis showed that the level of disease acceptance decreased with age (*p* = 0.01) [49]. The highest mean score for disease acceptance was reported in the age group of up to 30 years (34.41 points) and in the age group of 31–40 years (33.3 points). The lowest score was determined in the age groups of 41–50 years (29.63 points) and over 60 years (27.67 points).

Moreover, people who monitor blood pressure less often are more likely to be unadjusted. Blood pressure measurement is a common diagnostic and monitoring procedure, and accuracy is essential if patients are to receive the appropriate treatment and care in a timely manner. Accurate blood pressure measurement is, therefore, vital in the prevention and treatment of blood-pressure-related diseases. Additionally, in very ill patients, accurate measurement of blood pressure is essential for monitoring cardiovascular homeostasis and transmit tranquility to patients [50].

However, after the educational intervention, it was found that results are contrary to those of the previous moment, because participants who live alone have a greater possibility of being “unadjusted” in this subscale than those who live with more than two people. In the study by Yildiz and Erci, the authors found that most participants lived with their spouses and/or children, and that this fact significantly influenced the frequency of blood pressure assessment [2]. Regarding control of anxiety and levels of stress at home, these authors found that there were no statistically significant results among participants who lived alone or accompanied. This may be related to the results of our study, because, after educational nursing intervention patients, who assess blood pressure more frequently and live accompanied by family members may have lower levels of anxiety than patients living alone, who remain more carefree. Consequently, these factors may influence the adoption, or not, of healthy lifestyles and lead to an increased risk of cardiovascular accidents. While family support can be very important for these patients, in the case of negative family relationships, they can cause stress, affect mental health, and even cause physical symptoms [39].

In the fatalism subscale, prior to the educational intervention, participants who were at working age and who measured blood pressure less often had a higher possibility of being “unadjusted” in this subscale than retired users who checked blood pressure every day or once a week.

According to Hamer, Batty, Stamatakis, and Kivimaki, there is evidence of association, in some patients between hypertension and feelings of fatalism, such as exacerbation of prognosis and future life [51]. According to these authors, the association may be due to a direct effect of increased blood pressure, related to adverse effects of treatment, or consequences of labeling. Thus, individuals “labeled” as hypertensive may adopt a role of patient that can impair their quality of life.

According to Ojike et al., a person’s mental health status critically affects their ability to maintain a healthy lifestyle, seek early treatment for comorbidity conditions, or consistently adhere to treatment programs [31]. This nonadherence to therapeutic regimen constitutes a higher probability of blood pressure instability, a greater tendency to complications, and finally a reduction in quality of life [42,44].

Regarding the avoidance/denial subscale, before the educational intervention, it was found that participants at working age were more likely to be “unadjusted” in this subscale than retired users. In this regard Chen, Zhou, Liu, and Yu reported that feelings of denial are more frequent in people at working age, because chronic diseases can limit their performance [52].

In this context of acceptance of the disease, it is important that hypertensive patients understand that lowering blood pressure has benefits, such as those related to stroke (35–40%), acute myocardial infarction (20–25%), and heart failure (more than 50%) [4].

Educational interventions can create opportunities for patients to better understand their conditions and the role of surveillance consultations, as well as raise awareness of disease progression and complications. Through health education, wrong concepts that patients have about their treatment can be clarified, and health professionals can improve patient knowledge [5]. This was confirmed in the present study, where there were a considerable number of unadjusted patients at the first stage, who moved to the category of adjusted in the subscales of “despair/hopelessness”, “anxious concern”, “fatalism”, and “avoidance/denial” at the second stage, after the educational intervention.

Thus, educational interventions can positively modify patient beliefs, which in turn can lead to a change in their behavior, such as better adherence to a treatment suggested by the health professional and a possible effect on disease-related variables, such as a decrease in blood pressure values. In this context, the role of nurses in promoting self-care in patients with hypertension includes planning, managing, and evaluating nursing interventions to train the individual in lifestyle changes, to increase awareness of the potential complications of hypertension, and to observe behavioral changes after such instruction [2]. However, management of hypertension, as a lifelong disease, is a long challenge, which often requires the patient to practice self-care throughout life. As already mentioned, this is a factor that can cause ideological burden and increase the likelihood of patients presenting negative emotions, such as fear, anxiety, and depression [44].

According to the results obtained in our study, it seems evident that the educational nursing intervention carried out with hypertensive patients promoted their mental adjustment. Thus, we can say that the topics addressed/developed in the educational intervention (medication regimen, diet, and physical exercise; hypertension as a chronic disease and complications of hypertension; self-measurement of blood pressure and quality of life in hypertension; changes in food confection and physical exercise; self-surveillance of hypertension symptoms and non-pharmacological strategies for anxiety control) were effective in promoting mental adjustment. Regarding this aspect, Ho et al. implemented an educational intervention in hypertensive patients, providing oral and written information that included the definition of hypertension, its causes, cardiovascular risk factors, and control measures [53]. With this intervention, the authors significantly improved the patients’ knowledge.

In line with several authors, we believe that the use of telemedicine as a strategy to involve patients in self-management of hypertension can provide continuous monitoring and surveillance of these parameters by health professionals, identify symptoms early, and allow a prompt response to exacerbations of the disease [54,55,56]. In addition to these possibilities, it could help to improve patient knowledge, by watching small films about the adoption of healthy lifestyle habits, or by sending adequate information by health professionals, as an educational strategy.

One of the main limitations of our study was the value of the internal consistency of the “fighting spirit” subscale. Upon reducing the items of this subscale, from 22 to 15 items, a Cronbach alpha score of 0.611 was obtained (minimum value: 15 points; maximum value: 60 points; cutoff point: 37.5 points), and the value and percentage of unadjusted participants became 6 (1.8%) (M = 45.71; SD = 4.25; minimum: 35 points; maximum: 56 points). Thus, even with the elimination of seven items, and with the “best” alpha score achieved, the value of unadjusted was residual when compared to the other subscales, which may indicate that this subscale cannot be used for chronic diseases, without major changes in the scale itself. For this reason, we suggest a future study with the aim of verifying the reliability of this subscale when used in chronic diseases other than cancer.

Another limitation of our study was the potential effects of external factors or other interventions not related to the educational intervention of the present study, which may have influenced the responses of some participants.

In addition, the study schedule was another limitation. The period for substantial change in mental adjustment to the disease was relatively short. In other words, the period between the educational intervention and the second evaluation (1 month after the intervention) may not have been sufficient for a change in behaviors and consequent mental adjustment. Thus, it seems important to carry out a study with longer follow-up (e.g., 6 months or 1 year) and with multiple educational sessions, to ensure a more appropriate period for a relatively stable change of behavior.

## 6. Conclusions

In this study, we found that the educational intervention implemented in the participants who were mentally “unadjusted” to arterial hypertension had a positive effect on a decrease in the number of “unadjusted” classifications in the multiple categories of the MADS. We also found that patients who lived with other family members, who were in an active professional situation before the diagnosis of hypertension and who currently still were, under the age of 65 years, with a shorter time to diagnosis, and with a less regular measurement of blood pressure were more likely to be mentally unadjusted to arterial hypertension.

In patients with hypertension, there is a multiplicity of intrinsic and extrinsic factors that influence physical and mental dimensions and that change over time, through exposure to health/disease experiences. Thus, these factors should be considered in the design of health programs and interventions.

Educational intervention performed by the FN is important in helping the patient and their family in defining strategies to adjust to the disease, through mindfulness, offering the opportunity for face-to-face communication and teaching and assessing the person’s health situation. Thus, our study showed that educational intervention performed by the FN had a positive effect on the reduction of patients who were unadjusted to arterial hypertension.

Nurses need to be aware to negative emotions of the patient, such as despair, anguish, anxiety, stress, overvaluation of the disease, and denial, as well as value their own role in nonadherence to treatment. Thus, the recommendation is that attention be focused on the importance of evaluating and working on mental adjustment to the disease and consequent acceptance, as a possible mechanism via which negative emotions could be managed by hypertensive patients with the help of health professionals.

## Figures and Tables

**Figure 1 ijerph-19-00170-f001:**
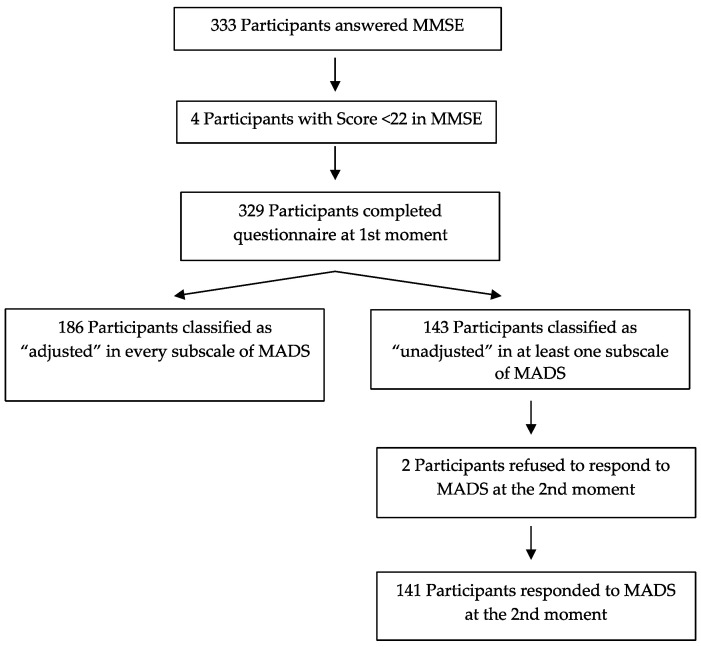
Flowchart of constitution of the samples under study.

**Table 1 ijerph-19-00170-t001:** Cutoff points for “adjusted/unadjusted” classification for subscales used.

MADS Subscales	No. of Items	Minimum	Maximum	Score for “Adjusted”	Score for “Unadjusted”
Fighting spirit	22	22	88	56–88	22–55
Despair/hopelessness	6	6	24	6–15	16–24
Anxious concern	9	9	36	9–23	24–36
Fatalism	9	9	36	9–23	24–36
Avoidance/denial	1	1	4	1–2	3–4

**Table 2 ijerph-19-00170-t002:** Structure of educational intervention applied to “unadjusted” participants.

MADS Subscales	Topics Addressed/Developed
Fighting spirit	Medication regimen, diet, and physical exercise
Avoidance/denial	Hypertension as chronic disease and complications of hypertension
Fatalism	Self-measurement of blood pressure and quality of life in hypertension
Despair/hopelessness	Changes in food confection and physical exercise
Anxious concern	Self-surveillance of hypertension symptoms and non-pharmacological strategies for anxiety control

**Table 3 ijerph-19-00170-t003:** Results of internal consistency (Cronbach’s alpha) obtained in our study and comparison with other studies.

	Our Study		Watson et al., (1988) (*n* = 235)	Schwartz et al., (1992) (*n* = 238)	Nordin et al., (1999) (*n* = 868)	Sá (2001) (*n* = 122)
MADS Subscales	First Assessment (*n* = 329)	Second Assessment (*n* = 141)				
Fighting spirit	0.25	*	0.84	0.78	0.81	0.86
Despair/hopelessness	0.92	0.92	0.79	0.83	0.78	0.83
Anxious concern	0.92	0.76	0.65	0.43	0.62	0.60
Fatalism	0.94	0.89	0.65	0.67	0.61	0.64
Avoidance/denial	n.a.	n.a.	n.a.	n.a.	n.a.	n.a.

n.a.—not applicable (number of items = 1); * not evaluated because all participants were classified as “adjusted” in this subscale in the first evaluation and did not participate in the second one.

**Table 4 ijerph-19-00170-t004:** Sociodemographic characterization of participants (*n* = 329).

Sociodemographic Variables	*n* (%)	Clinical Characterization	*n* (%)
*Gender*		*Professional Situation prior to disease*	
Masculine Feminine	136 (41.3) 193 (58.7)	Student	0 (0)
		Employed	150 (45.6)
		Unemployed	6 (1.8)
		Retired	173 (52.6)
*Marital Status*		*Current professional situation*	
Single	32 (9.7)	Student	0 (0)
Married/living with a partner	209 (63.5)	Employed	61 (18.5)
Divorced/separated	12 (3.6)	Unemployed	11 (3.3)
Widowed	76 (23.1)	Retired	257 (78.1)
*Academic qualifications*		*Age (years; M ± SD)*	70.8 ± 11.0
No schooling Primary school Middle school Secondary school Higher education	35 (10.6) 205 (62.3) 61 (18.5) 17 (5.2) 11 (3.3)	18–30	0 (0)
		31–40	2 (0.6)
		41–50	15 (4.6)
		51–60	40 (12.2)
		61–70	95 (28.9)
		71–80	107 (32.5)
		81–90	64 (19.5)
		>90	9 (1.8)
*Composition of the household*		*Number of people living with (M ± SD)*	2.1 ± 1.0
Does not live with anyone Lives with spouse/partner Lives with children/grandchildren Lives with mother/father-in-law/son/daughter-in-law Lives with brothers/sisters/mother/father	93 (28.3) 207 (62.9) 100 (30.4) 2 (0.6) 18 (5.4)	Lives alone	93 (28.3)
		Lives with 1 person	150 (45.6)
		Lives with 2 persons	50 (15.2)
		Lives with 3 persons	25 (7.6)
		Lives with 4 persons	9 (2.7)
		Lives with 5 persons	2 (0.6)
*Job*			
Member of the armed forces	1 (0.3)		
Experts in intellectual and scientific activities	9 (2.7)		
Mid-level technicians and professionals	3 (0.9)		
Administrative staff	17 (5.2)		
Personal protection, security, and sales workers	6 (1.8)		
Farmers and skilled workers in agriculture, fisheries and forests	6 (1.8)		
Skilled workers in industry, construction, and craftsmen	30 (9.1)		
Plant and machinery operators and assembly workers	54 (16.4)		
Unskilled workers	203 (61.7)		

**Table 5 ijerph-19-00170-t005:** Clinical characterization of participants (*n* = 329).

Clinical Variables	*n* (%)	Clinical Variables	*n* (%)
*Takes medication for hypertension*		*Site of blood pressure measurement*	
No Yes	1 (0.3) 328 (99.7)	Consultations	103 (31.3)
		Consultations and home	157 (47.7)
		Consultations, home and Pharmacy	14 (4.3)
		Consultations and Pharmacy	55 (16.7)
*How long has been medicated*		*Frequency of blood pressure measurement*	
For about 1 month For about 1 year For more than 1 year	1 (0.3) 20 (6.1) 307 (93.3)	Once a year	1 (0.3)
		Twice or three times a year	110 (33.4)
		Once a month	117 (35.6)
		Once a week	84 (25.5)
		Every day	17 (5.2)
*Going to Hypertension Surveillance Consultations*		*Disease duration (years; M ± SD)*	8.4 ± 6.0
No Yes	14 (4.3) 315 (95.7)	1–2	51 (15.5)
		3–5	90 (27.4)
		6–10	90 (27.4)
		11–15	61 (18.5)
		16–20	21 (6.4)
		>20	16 (4.9)
*Reason for Missing Hypertension Surveillance Consultation*			
Lack of time	1 (7.1)		
Little importance	0 (0)		
Lack of means	2 (14.3)		
Forgetfulness	11 (78.6)		
Other	0 (0)		

**Table 6 ijerph-19-00170-t006:** Characterization of “adjusted/unadjusted” classification for “despair/hopelessness” subscale.

	Stage 1			Stage 2		
	Unadjusted *n* (%)	Adjusted *n* (%)	OR; 95% CI	Unadjusted *n* (%)	Adjusted *n* (%)	OR; 95% CI
*Gender*						
Masculine (G. Ref)	27 (19.9)	109 (80.1)	1	15 (57.7)	11 (42.3)	1
Feminine	45 (23.3)	148 (76.7)	1.22; 0.72–2.10	32 (71.1)	13 (28.9)	1.81; 0.66–4.96
*Marital Status*						
Single	8 (25.0)	24 (75.0)	1.21; 0.47–3.12	6 (75.0)	2 (25.0)	1.39; 0.21–8.98
Married/living with a partner	45 (21.5)	164 (78.5)	0.99; 0.54–1.83	28 (63.6)	16 (36.4)	0.81; 0.26–2.54
Divorced/separated/widowed (G. Ref)	18 (20.7)	69 (79.3)	1	13 (68.4)	6 (31.6)	1
*Academic Qualifications*						
No schooling (G. Ref)	7 (20.0)	28 (80.0)	1	4 (57.1)	3 (42.9)	1
Primary school	38 (18.5)	167 (81.5)	0.91; 0.37; 2.40	26 (70.3)	11 (29.7)	1.77; 0.34–9.27
Middle school	16 (26.2)	45 (73.8)	1.42; 0.52; 3.89	9 (56.3)	7 (43.8)	0.96; 0.16–5.80
Secondary/higher education	11 (39.3)	17 (60.7)	2.59; 0.84–7.96	8 (72.7)	3 (27.3)	2.00; 0.27; 14.78
*Age (years)*						
Younger than 65	28 (32.2)	59 (67.8)	1.66; 0.83–3.31	21 (77.8)	6 (22.2)	2.80; 0.76–10.26
Between 65 and 79	26 (16.1)	135 (83.9)	0.67; 0.34; 1.31	16 (61.5)	10 (38.5)	1.28; 0.38–4.34
Aged 80 or older (G. Ref)	18 (22.2)	63 (77.8)	1	10 (55.6)	8 (44.4)	1
*Household (no. of people living with)*						
Lives alone (G. Ref)	17 (18.3)	76 (81.7)	1	12 (70.6)	5 (29.4)	1
Lives with 1 person	28 (18.7)	122 (81.3)	1.03; 0.53–2.00	16 (59.3)	11 (40.7)	0.61; 0.16–2.21
Lives with 2 persons	17 (34.0)	33 (66.0)	**2.30; 1.05–5.06**	13 (76.5)	4 (23.5)	1.35; 0.29–6.26
Lives with >2 persons	10 (27.8)	26 (72.2)	1.72: 0.70; 4.23	6 (60.0)	4 (40.0)	0.63; 0.12–3.22
*Professional situation prior to disease*						
Employed/unemployed	47 (30.1)	109 (69.9)	**2.55; 1.48–4.40**	33 (71.7)	13 (28.3)	2.00; 0.72–5.52
Retired (G. Ref)	25 (14.5)	148 (85.5)	1	14 (56.0)	11 (44.0)	1
*Current professional situation*						
Employed/unemployed	26 (36.1)	46 (63.9)	**2.59; 1.46–4.62**	19 (76.0)	6 (24.0)	2.04; 0.68–6.07
Retired (G. Ref)	46 (17.9)	211 (82.1)	1	28 (60.0)	18 (39.1)	1
*Going to Surveillance Consultations of hypertension*						
Yes	3 (21.4)	11 (78.6)	0.97; 0.26–3.58	2 (100.0)	0 (0.0)	n.a.
No (G. Ref)	69 (21.9)	246 (78.1)	1	45 (65.2)	24 (34.8)	n.a.
*Disease duration (years)*						
1–2	17 (33.3)	34 (66.7)	1.73; 0.82–3.66	12 (70.6)	5 (29.4)	1.37; 0.35–5.33
3–5	14 (15.6)	76 (84.4)	0.64; 0.30–1.34	10 (76.9)	3 (23.1)	1.91; 0.40–9.02
6–10	19 (21.1)	71 (78.9)	0.92; 0.46–1.85	11 (57.9)	8 (42.1)	0.79; 0.22–2.77
>11 (G. Ref)	22 (22.4)	76 (77.6)	1	14 (63.6)	8 (36.9)	1
*Frequency of blood pressure measurement*						
Once to 3 times a year	26 (23.4)	85 (76.6)	1.24; 0.64–2.39	18 (72.0)	7 (28.0)	2.10; 0.61–7.27
Once a month	26 (22.2)	91 (77.8)	1.16; 0.60–2.23	18 (69.2)	8 (30.8)	1.84; 0.55–6.19
Every day or once a week (G. Ref)	20 (19.8)	81 (80.2)	1	11 (55.0)	9 (45.0)	1

Significant results in bold; G. Ref: reference group; n.a.: not applicable because there were cells with value 0.

**Table 7 ijerph-19-00170-t007:** Characterization of “adjusted/unadjusted” classification for “anxious concern” subscale.

	Stage 1			Stage 2		
	Unadjusted *n* (%)	Adjusted *n* (%)	OR; 95% CI	Unadjusted *n* (%)	Adjusted *n* (%)	OR; 95% CI
*Gender*						
Masculine (G. Ref)	21 (15.4)	115 (84.6)	1	9 (45.0)	11 (55.0)	1
Feminine	36 (18.7)	157 (81.3)	1.26; 0.70–2.26	22 (61.1)	14 (38.9)	1.92; 0.63–5.81
*Marital status*						
Single	7 (21.9)	25 (78.1)	1.36; 0.50–3.73	7 (100)	0 (0.0)	n.a.
Married/living with a partner	35 (16.7)	174 (83.3)	0.98; 0.50–1.90	13 (38.2)	21 (61.8)	n.a.
Divorced/separated/widowed (G. Ref)	15 (17.0)	73 (83.0)	1	11 (73.3)	4 (26.7)	n.a.
*Academic qualifications*						
No schooling (G. Ref)	3 (8.6)	32 (91.4)	1	3 (100)	0 (0.0)	n.a.
Primary school	31 (15.1)	174 (84.9)	1.90; 0.55–6.59	16 (53.3)	14 (46.7)	n.a.
Middle school	13 (21.3)	48 (78.7)	2.89; 0.76–10.95	8 (61.5)	5 (38.5)	n.a.
Secondary/higher education	10 (35.7)	18 (64.3)	**5.93; 1.44–24.36**	4 (40.0)	6 (60.0)	n.a.
*Age (years)*						
Younger than 65	28 (32.2)	59 (67.8)	**3.36; 1.51–7.50**	13 (48.1)	14 (51.9)	0.93; 0.22–3.96
Between 65 and 79	19 (11.8)	142 (88.2)	095; 0.42–2.15	13 (68.4)	6 (31.6)	2.17; 0.45–10.44
Aged 80 or older (G. Ref)	10 (12.3)	71 (87.7)	1	5 (50.0)	5 (50.0)	1
*Household (no. of people living with)*						
Lives alone (G. Ref)	13 (14.0)	80 (86.0)	1	11 (84.6)	2 (15.4)	**9.90; 1.53–63.69**
Lives with 1 person	19 (12.7)	131 (87.3)	0.89; 0.42–1.91	10 (55.6)	8 (44.4)	2.25; 0.54–9.45
Lives with 2 persons	11 (22.0)	39 (78.0)	**1.74; 0.71–4.23**	5 (45.5)	6 (54.5)	1.50; 0.30–7.53
Lives with >2 persons	14 (38.9)	22 (61.1)	**3.92; 1.61–9.54**	5 (35.7)	9 (64.3)	1
*Professional situation prior to disease*						
Employed/unemployed	37 (23.7)	119 (76.3)	**2.38; 1.31–4.31**	18 (50.0)	18 (50.0)	1
Retired (G. Ref)	20 (11.6)	153 (88.4)	1	13 (65.0)	7 (35.0)	1.86; 0.60–5.73
*Current professional situation*						
Employed/unemployed	25 (34.7)	47 (65.3)	**3.74; 2.03–6.89**	13 (54.2)	11 (45.8)	0.92; 0.32–2.66
Retired (G. Ref)	32 (12.5)	225 (87.5)	1	18 (56.3)	14 (43.8)	1
*Going to Surveillance of hypertension*						
Yes	4 (28.6)	10 (71.4)	1.98; 0.60–6.54	2 (66.7)	1 (33.3)	1.66; 0.14–19.39
No	53 (16.8)	262 (83.2)	1	29 (54.7)	24 (45.3)	1
*Disease duration (years)*						
1–2	15 (29.4)	36 (70.6)	**2.99; 1.27–7.00**	10 (66.7)	5 (33.3)	2.00; 0.42–9.52
3–5	16 (17.8)	74 (82.2)	1.55; 0.69–3.49	7 (46.7)	8 (53.3)	0.88; 0.19–4.00
6–10	14 (15.6)	76 (84.4)	1.32; 0.58–3.03	8 (57.1)	6 (42.9)	1.33; 0.28; 6.28
>11 (G. Ref)	12 (12.2)	86 (87.8)	1	6 (50.0)	6 (50.0)	1
*Frequency of blood pressure measurement*						
Once to 3 times a year	29 (26.1)	82 (73.9)	**2.62; 1.26–5.48**	15 (53.6)	13 (46.4)	1
Once a month	16 (13.7)	101 (86.3)	1.18; 0.53–2.62	9 (56.3)	7 (43.8)	1.11; 0.32–3.83
Every day or once a week (G. Ref)	12 (11.9)	89 (88.1)	1	7 (58.3)	5 (41.7)	1.21; 0.31–4.76

Significant results in bold; G. Ref: reference group; n.a.: not applicable because there were cells with value 0.

**Table 8 ijerph-19-00170-t008:** Characterization of “adjusted/unadjusted” classification for “fatalism” subscale.

	Stage 1			Stage 2		
	Unadjusted *n* (%)	Adjusted *n* (%)	OR; 95% CI	Unadjusted *n* (%)	Adjusted *n* (%)	OR; 95% CI
*Gender*						
Masculine (G. Ref)	27 (19.9)	109 (80.1)	1	19 (70.4)	8 (29.6)	1
Feminine	40 (20.7)	153 (79.3)	1.06; 0.61–1.82	30 (75.0)	10 (25.0)	1.26; 0.42–3.77
*Marital status*						
Single (G. Ref)	5 (15.6)	27 (84.4)	1	3 (60.0)	2 (40.0)	1
Married/living with a partner	40 (19.1)	169 (80.9)	1.28; 0.46–3.53	30 (75.0)	10 (25.0)	2.00; 0.29–13.74
Divorced/separated/widowed	22 (25.0)	66 (75.0)	1.80; 0.62–5.24	16 (72.7)	6 (27.3)	1.78; 0.24–13.41
*Academic qualifications*						
No schooling (G. Ref)	7 (20.0)	28 (80.0)	1	4 (57.1)	3 (42.9)	1
Primary school	44 (21.5)	161 (78.5)	1.09; 0.45–2.67	33 (75.0)	11 (25.0)	2.25; 0.43–11.66
Middle school	10 (16.4)	51 (83.6)	0.78; 0.27–2.29	9 (90.0)	1 (10.0)	6.75; 0.53–86.56
Secondary/higher education	6 (21.4)	22 (78.6)	1.09; 0.32–3.71	3 (50.0)	3 (50.0)	0.75; 0.08–6.71
*Age (years)*						
Younger than 65 (G. Ref)	17 (19.5)	70 (80.5)	1	13 (76.5)	4 (23.5)	1.08; 0.24–4.90
Between 65 and 79	30 (18.6)	131 (81.4)	0.94; 0.49–1.83	21 (70.0)	9 (30.0)	0.78; 0.22–2.79
Aged 80 or older	20 (24.7)	61 (75.3)	1.35; 0.65–2.81	15 (75.0)	5 (25.0)	1
*Household (no. of people living with)*						
Lives alone (G. Ref)	22 (23.7)	71 (76.3)	1	15 (68.2)	7 (31.8)	1
Lives with 1 person	31 (20.7)	119 (79.3)	1.92; 0.67–5.54	25 (80.6)	6 (19.4)	3.21; 0.43–23.79
Lives with 2 persons	9 (18.0)	41 (82.0)	1.62; 0.58–4.50	7 (77.8)	2 (22.2)	6.25; 0.84–46.13
Lives with >2 persons	5 (13.9)	31 (86.1)	1.36; 0.42–4.47	2 (40.0)	3 (60.0)	5.25; 0.48–56.80
*Professional situation prior to disease*						
Employed/unemployed	39 (25.0)	117 (75.0)	**1.73; 1.00–2.97**	29 (74.4)	10 (25.6)	1.16; 0.39–4.35
Retired (G. Ref)	28 (16.2)	145 (83.8)	1	20 (71.4)	8 (28.6)	1
*Current professional situation*						
Employed/unemployed	16 (22.2)	56 (77.8)	1.15; 0.61–2.18	12 (75.0)	4 (25.0)	1.14; 0.31–4.12
Retired (G. Ref)	51 (19.8)	206 (80.2)	1	37 (72.5)	14 (27.5)	1
*Going to Surveillance Consultations of hypertension*						
Yes	5 (35.7)	9 (64.3)	2.27; 0.73–7.00	5 (100)	0 (0.0)	n.a.
No (G. Ref)	62 (19.7)	253 (80.3)	1	44 (71.0)	18 (29.0)	n.a.
*Disease duration (years)*						
1–2 (G. Ref)	12 (23.5)	39 (76.5)	1	8 (66.7)	4 (33.3)	1
3–5	16 (17.8)	74 (82.2)	0.70; 0.30–1.63	11 (68.8)	5 (31.3)	1.10; 0.22–5.44
6–10	13 (14.4)	77 (85.6)	0.54; 0.23–1.32	12 (92.3)	1 (7.7)	6.00; 0.56–63.98
>11	26 (26.5)	72 (73.5)	1.17; 0.53–2.58	18 (69.2)	8 (30.8)	1.13; 0.26–4.85
*Frequency of blood pressure measurement*						
Once to 3 times a year	16 (14.4)	95 (85.6)	**2.53; 1.31–4.91**	9 (56.3)	7 (43.8)	3.11; 0.86–11.29
Once a month	35 (29.9)	82 (70.1)	1.12; 0.53–2.37	28 (80.0)	7 (20.0)	2.33; 0.52–10.48
Every day or once a week (G. Ref)	16 (15.8)	85 (84.2)	1	12 (75.0)	4 (25.0)	1

Significant results in bold; G. Ref: reference group; n.a.: not applicable because there were cells with value 0.

**Table 9 ijerph-19-00170-t009:** Characterization of “adjusted/unadjusted” classification for “avoidance/denial” subscale.

	Stage 1			Stage 2		
	Unadjusted *n* (%)	Adjusted *n* (%)	OR; 95% CI	Unadjusted *n* (%)	Adjusted *n* (%)	OR; 95% CI
*Gender*						
Masculine (G. Ref)	19 (14.0)	117 (86.0)	1	9 (50.0)	9 (50.0)	1
Feminine	29 (15.0)	164 (85.0)	1.09; 0.58–2.04	18 (62.1)	11 (37.9)	1.64; 0.50–5.38
*Marital status*						
Single	9 (28.1)	23 (71.9)	2.48; 0.93–6.62	7 (77.8)	2 (22.2)	7.00; 0.97–50.57
Married/living with a partner	27 (12.9)	182 (87.1)	0.94; 0.45–1.95	16 (61.5)	10 (38.5)	3.20; 0.76–13.47
Divorced/separated/widowed (G. Ref)	12 (13.6)	76 (86.4)	1	4 (33.3)	8 (66.7)	1
*Academic qualifications*						
No schooling (G. Ref)	6 (17.1)	29 (82.9)	1	2 (33.3)	4 (66.7)	1
Primary school	23 (11.2)	182 (88.8)	0.61; 0.23–1.63	12 (54.5)	10 (45.5)	2.40; 0.36–15.94
Middle school	13 (21.3)	48 (78.7)	1.31; 0.45–3.82	9 (69.2)	4 (30.8)	4.50; 0.57–35.52
Secondary/higher education	6 (21.4)	22 (78.6)	1.32; 0.37; 4.64	4 (66.7)	2 (33.3)	4.00; 0.36–44.11
*Age (years)*						
Younger than 65	20 (23.0)	67 (77.0)	1.90; 0.85; 4.26	12 (60.0)	8 (40.0)	1.80; 0.41–7.96
Between 65 and 79	17 (10.6)	144 (89.4)	0.75; 0.33–1.69	10 (62.5)	6 (37.5)	2.00; 0.42–9.52
Aged 80 or older (G. Ref)	11 (13.6)	70 (86.4)	1	5 (45.5)	6 (54.5)	1
*Household (no of people living with)*						
Lives alone (G. Ref)	14 (15.1)	79 (84.9)	0.89; 0.31–2.52	7 (50.0)	7 (50.0)	0.50; 0.07–3.68
Lives with 1 person	17 (11.3)	133 (88.7)	0.64; 0.23–1.76	11 (68.8)	5 (31.3)	1.10; 0.15–8.13
Lives with 2 persons	11 (22.0)	39 (78.0)	1.41; 0.47–4.25	5 (45.5)	6 (54.5)	0.42; 0.05–3.31
Lives with >2 persons (G. Ref)	6 (16.7)	30 (83.3)	1	4 (66.7)	2 (33.3)	1
*Professional situation prior to disease*						
Employed/unemployed	25 (16.0)	131 (84.0)	1.25; 0.67–2.30	16 (66.7)	8 (33.3)	2.18; 0.67–7.09
Retired (G. Ref)	23 (13.3)	150 (86.7)	1	11 (47.8)	12 (52.2)	1
*Current professional situation*						
Employed/unemployed	16 (22.2)	56 (77.8)	**2.00; 1.03–3.92**	8 (50.0)	8 (50.0)	1
Retired (G. Ref)	32 (12.5)	225 (87.5)	1	19 (61.3)	12 (38.7)	1.58; 0.47–5.35
*Going to Surveillance Consultations of hypertension*						
Yes	3 (21.4)	11 (78.6)	1.64; 0.44–6.10	2 (66.7)	1 (33.3)	1.52; 0.13–18.03
No (G. Ref)	45 (14.3)	270 (85.7)	1	25 (56.8)	19 (43.2)	1
*Duration of disease (years)*						
1–2	11 (21.6)	40 (78.4)	2.42; 0.95–6.16	6 (54.5)	5 (45.5)	0.80; 0.14–4.53
3–5	15 (16.7)	75 (83.3)	1.76; 0.75–4.15	9 (60.0)	6 (40.0)	1.00; 0.20–5.12
6–10	12 (13.3)	78 (86.7)	1.35; 0.55–3.31	6 (54.5)	5 (45.5)	0.80; 0.14–4.53
>11 (G. Ref)	10 (10.2)	88 (89.8)	1	6 (60.0)	4 (40.0)	1
*Frequency of blood pressure measurement*						
Once to 3 times a year	26 (23.4)	85 (76.6)	2.07; 0.99–4.29	12 (48.0)	13 (53.0)	0.79; 0.21–3.03
Once a month	9 (7.7)	108 (92.3)	0.56; 0.23–1.38	8 (88.9)	1 (11.1)	6.86; 0.66–71.71
Every day or once a week (G. Ref)	13 (12.9)	88 (87.1)	1	7 (53.8)	6 (46.2)	1

Significant results in bold; G. Ref: reference group.

**Table 10 ijerph-19-00170-t010:** Results of comparative analysis of MADS between moment 1 and moment 2 using classification “adjusted/unadjusted” by subscale.

MADS Subscales	Classification	Stage 1		Stage 2	
		*n* (%)	M ± SD	*n* (%)	M ± SD
Fighting spirit	Adjusted	329 (100)	64.1 ± 4.4	n.a.	-
	Unadjusted	0 (0)	-	n.a.	-
Despair/hopelessness	Adjusted	257 (78.1)	8.2 ± 1.7	24 (33.8)	8.1 ± 1.4
	Unadjusted	72 (21.9)	18.9 ± 1.7	47 (66.2)	18.3 ± 1.5
Anxious concern	Adjusted	272 (82.7)	11.4 ± 1.4	25 (44.6)	19.5 ± 1.9
	Unadjusted	57 (17.3)	28.8 ± 1.9	31 (55.4)	24.8 ± 0.9
Fatalism	Adjusted	262 (79.6)	11.6 ± 2.1	18 (26.9)	11.2 ± 1.5
	Unadjusted	67 (20.4)	27.5 ± 1.7	49 (73.1)	26.8 ± 1.8
Avoidance/denial	Adjusted	281 (85.4)	1.4 ± 0.5	20 (42.6)	1.5 ± 0.5
	Unadjusted	48 (14.6)	3.2 ± 0.4	27 (57.4)	3.3 ± 0.4

n.a.: not applicable because there were no “unadjusted” participants at stage 1. Note: one participant dropped out the study at stage 2 after being classified as “unadjusted” at stage 1. This participant was unadjusted in all subscales, except for “fighting spirit” and “fatalism”.

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
