# Peer review of "Effect of an Educational Nursing Intervention on the Mental Adjustment of Patients with Chronic Arterial Hypertension: An Interventional Study"

_ijerph, 2021, doi:10.3390/ijerph19010170_

Round 1
Reviewer 1 Report
The manuscript addresses an extremely important public health problem not only in Portugal but also in all developed countries.
As this scale was never used in hypertensive patients and to ensure its internal consistency, the Cronbach’s alpha calculation was performed - is this enough to used in hypertensive patients?
Overly elaborate description of sociodemographic characteristics at the expense of survey results.
Table 4 is too large and therefore unreadable.
There was too much focus on sociodemographic variables and their impact on adjustment, rather than on the outcome of the intervention-the impact of education on adjustment in illness.
Too little about the effectiveness of interventions.
It would be interesting to address telemedicine as a support for patients and their families in the fight against chronic disease.
Author Response
Dear Reviewer, we are very grateful for your careful review of our article. Below we describe our response to each of your comments, suggestions, or questions:
1. The manuscript addresses an extremely important public health problem not only in Portugal but also in all developed countries.
R: Thank you very much for your comment. It really is a problem that worries health professionals in developed countries.
2. As this scale was never used in hypertensive patients and to ensure its internal consistency, the Cronbach’s alpha calculation was performed - is this enough to used in hypertensive patients?
R: Thank you very much for your question. The scale used in the study has been previously validated for the Portuguese population and for other chronic diseases, in addition to cancer. For us, the alpha calculation seemed to be sufficient to guarantee the reliability of the use of the scale.
3. Overly elaborate description of sociodemographic characteristics at the expense of survey results.
R: Thank you very much for your comment. In this study, we chose to carry out a more exhaustive approach to the sociodemographic characteristics of the participants, as it was an opportunity to better understand this population and see how these characteristics can influence the mental adjustment of patients with hypertension. Based on our clinical experience, we believe that these variables are important for mental adjustment and for planning the educational intervention to be implemented. We crossed these variables with mental adjustment according to one of the objectives of the study, so that it would be possible to better understand the profile of these patients and alert health professionals to the need for intervention in people with the characteristics with statistically significant results.
4. Table 4 is too large and therefore unreadable.
R: Thank you very much for your suggestion. We decided to divide Table 4 into two different tables (one relating to sociodemographic characteristics and the other relating to clinical characteristics).
5. There was too much focus on sociodemographic variables and their impact on adjustment, rather than on the outcome of the intervention-the impact of education on adjustment in illness.
R: Thank you very much for your comment. We added a paragraph to the discussion about the effectiveness of educational intervention.
6. Too little about the effectiveness of interventions.
R: Thank you very much for your comment. We added a paragraph to the discussion about the effectiveness of educational intervention.
7. It would be interesting to address telemedicine as a support for patients and their families in the fight against chronic disease.
R: As suggested, we have added the following text to the discussion:
“Based on several authors, we believe that the use of telemedicine as a strategy to involve patients in self-management of hypertension can provide continuous monitoring and surveillance of these parameters by health professionals, identify symptoms early and respond promptly to exacerbations of the disease [54-56]. In addition to these possibilities, it could also help to improve patients' knowledge, by watching small films about the adoption of healthy lifestyle habits, or by sending adequate information by health professionals, as an educational strategy.”
Reviewer 2 Report
This study describes a quantitative study investigating the effect of an educational nursing intervention on mental adjustment of patients with chronic hypertension. While many people may be unaware that they have hypertension, its global prevalence is already highly recognized with the number of adults suffering from hypertension nearly doubled in the past thirty years. The current study, adopting a validated tool established for cancer intervention, is expected to be of interest to health professionals and researcher. In addition, the current study is also an extended collection of recent relevant studies published in IJERPH, as also adequately cited in the manuscript. Educational health promotion interventions have promising potential and is worth investigation. This study uses a validated tool, which was established for cancer disease, on the important issues regarding hypertension. The study is considered of significance and has identified several factors for consideration in the design of health programmes and interventions.
The study has described in detail the design, sampling, data acquisition and methodology. The limitation regarding internal consistency was also acknowledged and illustrated with relevant manipulations of Cronbach’s alpha. Overall, the study was well described. The results from the study, whether consistent with or in contrary to literature findings, were discussed with relevance to reported studies and reference. I have a few suggestions as follows:
1.
Ln 36-37: For the statement “According to the World Health Organization…affects nearly 1 billion people worldwide”, it merits citation of the official data source. May also consider more accurate delivery of information which reports an estimation of 1.28 billion adults aged 30-79 years having hypertension:
https://www.who.int/news-room/fact-sheets/detail/hypertension
2.
Would authors consider citation for the statement on significant increase in average life expectancy in ln 66-67.
3.
Ln 575-576: appreciate the perspective from the authors, would the probable reasons also include young people having health literacy or knowing more about the consequence of the chronic condition.
4.
Page 14, Table 6, there is a note talking about one participant dropped out the study at moment 2, therefore among the 2nd, 3rd and 5th subscales, the total number n at moment 2 is one less from that at moment 1, respectively. For example, regarding “anxious concern”, n at moment 1 = 57 and n at moment 2 = 25 + 31 = 56. This applies to all other subscales except for the subscale “fatalism”, n at moment 1 = 67 while n at moment 2 = 49 + 18 = 67, can authors clarify or have updates on this observation.
5.
Ln 61-64: the authors mention that there are no studies addressing the implementation of educational intervention on mental adjustment to arterial hypertension, it is worth mentioning related interventions on other diseases like cancer for readers’ reference at this point, although there are relevant citations in latter part of the manuscript.
6.
Ln 71-72: should “arterial hypertension has gained…” be separate statement from the preceding description?
7.
Ln 135: suggest replacing “discouragement/hopelessness” by “despair/hopelessness”, which is used in other parts of the manuscript, including the table and for the evaluation tool.
8.
For the flowchart diagram on page 5 (Figure 1), in the figure, regarding the final box, the word “to” is missing after “responded”.
9.
Page 9, Table 4, please check the correct alignment of the respective percentages of the last three items under the category “Job”. Similarly for the percentages under the category “Site of blood pressure measurement”.
10.
Page 10, Table 5A, the name of one of the categories in the table was written in Portuguese? Should it be “Current Professional Situation”?
11.
Ln 387-389, I do not quite understand if the statement after “because” is actually explaining the contrary result of just a mere description of the observation. If the latter, should it be mentioned as the current state which is like a cause-and-effect relationship?
12.
Ln 407: the name of the variable should be “current professional situation” rather than “professional situation”.
13.
Ln 457: delete the extra dot after subscale; ln 639: it seems a word is missing after “its”.
14.
Ln 641: suggest replacing “hopelessness” with “despair/hopelessness”, similarly “avoidance/denial” with “denial”, as for consistent naming throughout the manuscript.
Author Response
Dear Reviewer, we are very grateful for your careful review of our article. Below we describe our response to each of your comments, suggestions, or questions:
This study describes a quantitative study investigating the effect of an educational nursing intervention on mental adjustment of patients with chronic hypertension. While many people may be unaware that they have hypertension, its global prevalence is already highly recognized with the number of adults suffering from hypertension nearly doubled in the past thirty years. The current study, adopting a validated tool established for cancer intervention, is expected to be of interest to health professionals and researcher. In addition, the current study is also an extended collection of recent relevant studies published in IJERPH, as also adequately cited in the manuscript. Educational health promotion interventions have promising potential and is worth investigation. This study uses a validated tool, which was established for cancer disease, on the important issues regarding hypertension. The study is considered of significance and has identified several factors for consideration in the design of health programmes and interventions.
The study has described in detail the design, sampling, data acquisition and methodology. The limitation regarding internal consistency was also acknowledged and illustrated with relevant manipulations of Cronbach’s alpha. Overall, the study was well described. The results from the study, whether consistent with or in contrary to literature findings, were discussed with relevance to reported studies and reference. I have a few suggestions as follows:
R: Thank you very much for the excellent summary you made of our article. We are very pleased that you found our study interesting, and we appreciate your compliments.
1. Ln 36-37: For the statement “According to the World Health Organization…affects nearly 1 billion people worldwide”, it merits citation of the official data source. May also consider more accurate delivery of information which reports an estimation of 1.28 billion adults aged 30-79 years having hypertension:
R: Thank you very much for the suggestion. We've improved the text and added the bibliographic reference.
2. Would authors consider citation for the statement on significant increase in average life expectancy in ln 66-67.
R: As suggested, a reference related to the statement made has been introduced.
3. Ln 575-576: appreciate the perspective from the authors, would the probable reasons also include young people having health literacy or knowing more about the consequence of the chronic condition.
R: Thank you very much for your comment. We did not assess knowledge about healthy health habits (health literacy), so we were unable to make a relationship between the remaining variables and patients' knowledge.
4. Page 14, Table 6, there is a note talking about one participant dropped out the study at moment 2, therefore among the 2nd, 3rd and 5th subscales, the total number n at moment 2 is one less from that at moment 1, respectively. For example, regarding “anxious concern”, n at moment 1 = 57 and n at moment 2 = 25 + 31 = 56. This applies to all other subscales except for the subscale “fatalism”, n at moment 1 = 67 while n at moment 2 = 49 + 18 = 67, can authors clarify or have updates on this observation.
R: Thank you very much for your question, which is very important. We went to check and the participant who dropped out was adjusted on the "Fatalism" and “Fighting Spirit” subscales and unadjusted on the other subscales. That is why there is a difference in the number of participants in the different subscales of the second moment. We have added this information in a footnote to the Table 6.
5. Ln 61-64: the authors mention that there are no studies addressing the implementation of educational intervention on mental adjustment to arterial hypertension, it is worth mentioning related interventions on other diseases like cancer for readers’ reference at this point, although there are relevant citations in latter part of the manuscript.
R: Thanks for the suggestion. We introduce a little reference to nurses' interventions, as we explain it better in the Discussion of results section.
6. Ln 71-72: should “arterial hypertension has gained…” be separate statement from the preceding description?
R: As suggested, we started a new sentence from "arterial hypertension..."
7. Ln 135: suggest replacing “discouragement/hopelessness” by “despair/hopelessness”, which is used in other parts of the manuscript, including the table and for the evaluation tool.
R: As suggested, we made the change in the text.
8. For the flowchart diagram on page 5 (Figure 1), in the figure, regarding the final box, the word “to” is missing after “responded”.
R: As suggested, we made the change in Figure 1.
9. Page 9, Table 4, please check the correct alignment of the respective percentages of the last three items under the category “Job”. Similarly for the percentages under the category “Site of blood pressure measurement”.
R: As suggested, we changed the alignment of percentages. We also decided to split the table into two (one for sociodemographic characteristics and one for clinical characteristics), as suggested by another reviewer.
10. Page 10, Table 5A, the name of one of the categories in the table was written in Portuguese? Should it be “Current Professional Situation”?
R: The suggested change was performed in Table 5A.
11. Ln 387-389, I do not quite understand if the statement after “because” is actually explaining the contrary result of just a mere description of the observation. If the latter, should it be mentioned as the current state which is like a cause-and-effect relationship?
R: The text was changed to make it clearer, as suggested.
12. Ln 407: the name of the variable should be “current professional situation” rather than “professional situation”.
R: The suggested change was performed in the text.
13. Ln 457: delete the extra dot after subscale; ln 639: it seems a word is missing after “its”.
R: The suggested changes was performed in the text.
14. Ln 641: suggest replacing “hopelessness” with “despair/hopelessness”, similarly “avoidance/denial” with “denial”, as for consistent naming throughout the manuscript.
R: The suggested changes was performed in the text.